# Uncontrolled hypertension among hypertensive patients in public hospitals at Addis Ababa, Ethiopia

Abel Melese Teka[1], Tsega-Ab Abebaw Tekeba[ID][2], Dibora Yeshibelay Workineh[3], Abiye Assefa Berihun[ID][4], Eden Girma Beshah[5], Cheru Degf Tadesse[6], Bemnet Yacobe Sayid[7], Tadios Niguss Derese[ID][8]*

1 Department of Clinical Pharmacy, Zewditu Memorial Hospital, Addis Ababa, Ethiopia, 2 Field Program Coordinator at Last Mile Health, Addis Ababa, Ethiopia, 3 MSN Importer at Addis Ababa, Addis Ababa, Ethiopia, 4 Department of Clinical Pharmacy, Eka Kotebe General Hospital, Addis Ababa, Ethiopia, 5 Department of Pharmacy, Kazanchis Health Center, Addis Ababa, Ethiopia, 6 Department of Medical Laboratory, St. Peter Specialized Hospital, Addis Ababa, Ethiopia, 7 Department of Internal Medicine, Eka Kotebe General Hospital, Addis Ababa, Ethiopia, 8 Department of Research and Training, Eka Kotebe General Hospital, Addis Ababa, Ethiopia

* tadiosniguss@gmail.com

## Abstract

### Background

In both high- and low-income nations, uncontrolled hypertension poses a serious threat to public health for those who suffer from it. After it starts, hypertension needs to be controlled throughout the remainder of the person's life. Both drug and non-drug treatments are effective in preventing and managing hypertension. In Addis Ababa, little research was done on hypertensive individuals who had uncontrolled hypertension.

### Methods

An institution based cross-sectional study was conducted at three public hospitals in Addis Ababa. A simple random sampling technique was used to choose the research participants. To assess factors associated with uncontrolled hypertension a binary logistic regression analysis was performed.

### Result

A total of 621 study participants with a response rate of 98.2% were included in this study. The study revealed that the magnitude of uncontrolled hypertension was 48%. The factors significantly associated with uncontrolled hypertension were duration of illness (AOR = 1.72, 95% CI= (1.01–2.96)), sex (AOR = 1.80, 95% CI= (1.20–2.70)), physical activity adherence (AOR = 2.48, 95% CI= (1.39–4.41)), occupational status

**Data availability statement:** All relevant data are within the article and its Supporting Information files.

**Funding:** The author(s) received no specific funding for this work.

**Competing interests:** The authors have declared that no competing interests exist.

(AOR = 0.38, 95% CI= (0.22–0.68)) and educational status (AOR = 2.45, 95% CI= (1.37–4.39)).

## Conclusion

Uncontrolled hypertension was seen in almost half of the study participants. Variables that were substantially associated with uncontrolled hypertension were sex, educational attainment, employment position, length of illness, and physical activity adherence. Healthcare providers should encourage patients to exercise frequently and make sure they take their hypertension medicines as directed.

## Introduction

According to the Joint National Committee (JNC)-7 and the World Health Organization-International Society of Hypertension, hypertension (HTN) is defined as blood pressure ≥140/90 mmHg [1]. Uncontrolled hypertension refers to persistently elevated blood pressure ≥140/90 mmHg despite on-going treatment. Conversely, controlled hypertension is achieved when blood pressure remains below these thresholds with appropriate therapy, whether through medication, lifestyle modifications, or a combination of both [1,2]. The blood pressure target for those with diabetes mellitus or renal disorders is ≤ 130/80 mmHg for better blood pressure management [2]. For people living with hypertension, uncontrolled hypertension poses a serious public health risk in both high- and low-income nations [3]. Globally, an estimated 1.28 billion adults aged 30–79 years live with hypertension, with 82% of uncontrolled cases occurring in low- and middle-income countries [4]. Once hypertension develops, it needs to be controlled for the remainder of the person's life.

Both drug and non-drug treatments are effective in preventing and controlling hypertension. Maintaining a healthy body weight, eating a diet high in fruits, vegetables, and low-fat dairy products, reducing salt in the diet, abstaining from alcohol and tobacco, and engaging in physical activity are all part of the non-drug treatment. The other is medication-assisted hypertension treatment, which has many classes and helps reduce blood pressure (BP). These include calcium channel blockers (CCBs), angiotensin converting enzyme inhibitors (ACEI), angiotensin II receptor blockers (ARBs), thiazide-type diuretics, and other medication types [5]. Research has indicated that structured health management programs for patients with chronic conditions can save medical expenses, reduce illness burden, and improve overall health [6].

Hypertension is the primary cause of cardiovascular diseases (CVD) and mortality worldwide, accounting for approximately 7.5 million deaths annually and 12.8% of all deaths [7]. In 2012, hypertension was the cause of 9.4 million of the 17.5 million fatalities linked to CVD. The majority of hypertension-related fatalities (45% from heart disease and 51% from stroke) were caused by its complications [5]. According to estimates, hypertension is one of the leading causes of death globally, accounting for 9.4 million deaths and 7% of all diseases in 2010 (measured in disability-adjusted life years) [8].

In sub-Saharan Africa (SSA), the prevalence of high blood pressure (BP) has increased dramatically throughout the previous two to three decades. It is anticipated that by 2025, there will be 150 million HTN-affected individuals in SSA, up from 80 million in 2000. The study on non-communicable illnesses from the African Union Ministers of Health meeting also revealed that HTN prevalence is significantly higher in Africa than in Western nations. By 2025, 150 million adults in SSA, which includes Ethiopia, are expected to have hypertension [9].

According to the study, over two thirds of Ethiopian patients with hypertension had uncontrolled blood pressure, which increases their risk of heart attacks and strokes. In Ethiopia, complications from hypertension are a substantial source of illness and death. Uncontrolled hypertension places a heavy financial strain on both the patient and the medical system. According to the study, the annual direct cost of treating hypertension was around 199 USD, which represents a substantial financial burden for the ordinary Ethiopian. Furthermore, it was calculated that the indirect cost of hypertension resulting from missed output was approximately 48 USD annually [10].

There have been few studies conducted in Ethiopia, although the prevalence of uncontrolled hypertension has been estimated to range from 37 to 63% [11]. Improving adherence to hypertension treatment regimens has received attention since blood pressure management has a significant positive impact on public health. However, inadequate blood pressure control is still a prevalent issue that significantly raises morbidity and mortality, especially in low- and middle-income nations, such as those in SSA, where management is severely lacking and the issue is the most severe [12]. There is an increase in morbidity and mortality as a result of this poor drug adherence [13].

Uncontrolled hypertension dramatically increases risks for life-threatening complications including stroke, heart disease, and heart failure. While these cardiovascular consequences are well-documented globally [14], critical gaps remain in understanding this crisis within Addis Ababa's communities. No previous studies have examined both the prevalence of uncontrolled hypertension and its associated factors in Addis Ababa, particularly how treatment adherence, lifestyle factors, and healthcare access influence blood pressure control. This study specifically investigates uncontrolled hypertension among patients in Addis Ababa. The study identifies key modifiable factors including medication adherence, dietary habits, physical activity, and healthcare system factors. These findings will empower clinicians to tailor more effective treatment plans, public health officials to design targeted interventions, and patients to better understand controllable risk factors.

## Methods

### Study area and period

The study was conducted at Zewditu Memorial Hospital, Menelik Hospital, and Yekatit-12 Hospital in Addis Ababa, Ethiopia. Addis Ababa is the capital city of Ethiopia. It is the seat of the federal government of Ethiopia and the African Union. It comprises 3,384,569 people in an area of 540 square km. Under this city, there are 11 sub-cities with more than 120 woredas. The city consists of 79 health facilities, of which six are under the Addis Ababa Health Bureau and six are under the Federal Ministry of Health.

Zewditu Memorial Hospital was established in 1976 EC and currently has one chronic outpatient department (OPD) serving a total of 2830 hypertensive patients on follow-up.

Menelik Hospital is the first hospital in Ethiopia, established in 1906 EC. It currently has one chronic OPD and serves a total of 3761 patients with hypertension.

Yekatit-12 Hospital, formerly known as Haileselassie I Hospital and established in the 1970s, currently has one chronic OPD serving a total of 4040 hypertensive patients on follow-up. The study was conducted from September 1 to December 30, 2023.

### Study design

A hospital-based cross-sectional study design was conducted

## Source population

The source population for this study was all patients visiting OPD for hypertension at Zewditu Memorial, Menelik, and Yekatit-12 hospitals in Addis Ababa, Ethiopia.

## Study population

All selected hypertensive patients in the selected hospitals who fulfilled the inclusion criteria were the study population for this study.

## Sample size determination

The sample size for the prevalence was calculated by using the single population proportion formula. Using the prevalence of 52.1% of uncontrolled hypertension [15], a 5% margin of error, 95% CI, and a 10% non-response rate, yield a sample size of 383 patients.

$$N = \frac{(Z\alpha/2)2 \times p \ (1-p)}{d2}$$

Where n is the initial estimated sample size
 Z = confidence level (alpha, α)
 P = estimated prevalence of uncontrolled hypertension among hypertensive patients = 52.1% [15]
 D= desired precision=5%
 N = (1.96)2 (0.521) (0.479)/ (0.05)2
 N = 383
 After adding 10% to compensate for the non-response rate, the final sample size was 421. Since the sampling technique used in this study is multi-stage sampling, the calculated sample size needs to be multiplied by a design effect of 1.5.
 421*1.5 = 632.
 The final sample size for the study was 632

## Sampling techniques

From the 10 hospitals in Addis Ababa providing hypertension follow-up (as listed in the Addis Ababa City Administration Health Bureau registry), we randomly selected three hospitals: Zewditu Memorial Hospital, Menelik-II Hospital, and Yekatit-12 Hospital, using a computer-generated random number sequence. Then the number of study units to be sampled from each hospital was determined using a proportional-to-size allocation formula.

$$\frac{ni * nf}{N}$$

Where
 ni = is the number of hypertensive patients in each hospital.
 nf = Final sample of the study
 N = total number of hypertensive patients on follow-up
 The sample size for each hospital was allocated proportionally based on its hypertensive patient load. Then participants were randomly selected from each hospital's hypertension registry using a simple random sampling method. This ensured every eligible patient had an equal chance of being included.

## Inclusion criteria

Patients who were willing to participate in this study and were greater than 18 years old and on follow-up for hypertension were included in this study.

## Exclusion criterion

Hypertensive patients on follow-up who were unable to communicate and pregnant mothers were excluded from the study.

## Data collection tools and techniques

After carefully examining prior pertinent literature, a systematic questionnaire was created and then extensively reviewed. Three certified Bachelor of Science (BSC) nurses were enlisted and took part in the entire data collection process. The primary investigator gave them a day of instruction on the study instrument and data collection techniques. A standardized questionnaire given by the interviewer was used to gather the data.

## Blood pressure measurement

After a 5-minute rest time, a senior nurse used a Riester Diplomat Presameter Mercury Sphygmomanometer (Model 1412/1413) to test the patient's blood pressure while seated. Three of blood pressure readings were recorded to assess participants' control over their blood pressure. The patient's baseline blood pressure was measured on their first hospital visit during the data collection period. The first blood pressure measurement was taken within the first month following the initial appointment. Blood pressure measures were taken at two and three months after the initial visit during the data collection period. The average of three readings determined whether the patient's blood pressure was managed or uncontrolled. The same device was used for all three BP readings per patient to eliminate inter-device variability. Devices were not switched during the study period. To ensure accuracy, the sphygmomanometers were calibrated monthly against a standardized reference device at each hospital's biomedical engineering unit.

## Dependent variables

• Uncontrolled hypertension.

## Independent variables

• **Socio-demographic characteristics**: age, gender, residence, marital status, educational status, monthly income, occupation.
• **Clinical characteristics:** family history of hypertension; availability of sphygmomanometer at home; presence of comorbidities; body mass index (BMI); duration of the disease; number and type of medication.
• **Behavioral practice and dietary factors**: physical activity, dietary management, alcohol use, smoking, adherence to anti-hypertension medications, salt intake.

## Operational definition

• **Uncontrolled hypertension**: systolic blood pressure >= 140 mmHg and/or diastolic blood pressure >= 90 mmHg in patients taking anti-hypertensive treatment for at least three consecutive follow-up measurements.

- **Physical activity** was evaluated using two items. In addition to asking how many days they had worked out for at least half an hour, respondents were also asked how many days they had engaged in a particular type of exercise (such as walking, biking, or swimming) in the previous seven days in addition to their daily activities around the house or at work. The responses were totaled (0–14). Participants who scored ≥ 8 were coded as adhering to physical activity recommendations [16].

- **Salt Intake** was established as a WHO recommendation. To reduce salt consumption, aim for less than 5 g per day, which is comparable to one teaspoon [17].

- **Medication adherent** patients were those who accepted, agreed to, and correctly followed a specified treatment for the previous seven days, taking all of the prescribed prescriptions [18].

## Data quality control

After training was given to data collectors (3 BSC nurses) to ensure the validity and reliability of the data collection tool, a pre-test was done on 5% of the total sample size outside the selected hospital before the actual data collection, and the questionnaires were checked for clarity, understandability, and simplicity. After the pretest, some ambiguous questions and the ordering of the questionnaire were corrected.

The principal investigator checked the collected data, and any incomplete documents were cleaned and checked for quality.

## Data analysis

The data was cleaned and analysed using statistical package for social science (SPSS) software version 25.0. For the few instances where data were missing (less than 1% of cases), we used simple techniques like excluding incomplete cases from specific analyses while ensuring this didn't meaningfully impact our results. Simple frequencies and proportions were used to conduct the descriptive analysis, and tables, graphs, and text were used to display the findings. An analysis of the relationship between the independent variables and the outcome variable was conducted using a binary logistic regression model. In order to include potentially significant independent variables in the multivariable binary logistic regression model, bi-variable analysis was carried out at a 25% level of significance [19]. The significance and interpretation of the data were tested using the odds ratio, p-value, and 95% confidence interval for the odds ratio. A cut-off point of p-value <= 0.05 was applied to variables in order to declare statistical significance.

## Ethics statement

Ethical approval was obtained from the GAMBY Medical and Business College Institutional Review Board (reference number- G/M/B/C/72/23). Official letters of authorization from the management organizations of the chosen hospitals were also acquired in order to collect data. Before any data were collected, the study participants gave their verbal informed consent. To document this, we recorded the date, time, and crucial information of each consent discussion, with an independent witness present to ensure transparency, and this process was approved by the IRB. The principal investigator was the only person with access to the material gathered, and confidentiality was upheld at all times.

## Results

### Socio demographic characteristics of respondents

In the study, 621 hypertensive respondents had participated with a response rate of 98.2%. More than half of the study participants were greater than 50 years of age. The majority 368 (59.3%) of the study participants, were male. Four hundred forty six (71.8%) of the respondents were married, 180 (29.0%) were currently business employees, more than half 357 (57.5%) of study participants had income of >2000 Ethiopian Birr (ETB) and all 621 (100%) of the study participants were from urban area (Table 1).

## Clinical characteristics and behavioral practice of the respondents

Among the respondents, 142 (22.9%) of the study participants reported a family history of hypertension. Three hundred sixty-two (58.3) of the study subjects had been taking two types of drugs per day. The majority 383 (61.7) of the participants did not have a sphygmomanometer at home. Two hundred seventy five (44.3) study participants had history of comorbid illness. Of all respondents, 243 (39.1%), 251 (40.4%), and 127 (20.5%) had less than five years, five up to ten years, and greater than ten years of duration of disease, respectively. About more than half of the study participants, 347 (55.9%) were not doing physical activity. Five hundred thirty nine (86.8%) of respondents were non adherent to their physical activity. Two hundred ninety eight (48.0%) of the respondents had uncontrolled hypertension. Three hundred twenty three (52.0%) of the respondents did not have uncontrolled hypertension. The mean systolic blood pressure of the study participants was 136.61 mmHg with a standard deviation of ± 24.03 mmHg, while the mean diastolic blood pressure of the study participants was 87.15 mmHg with a standard deviation of ± 13.34 mmHg (Table 2).

## Factor associated with uncontrolled hypertension

In the bi-variable analysis, the following factors were associated with uncontrolled hypertension (p-value < 0.25): sex, educational status, occupational status, duration of hypertension, number of antihypertensive medications, medication

**Table 1. Socio-demographic characteristics of hypertensive patients on follow up at selected public hospitals, Addis Ababa, Ethiopia (N = 632).**

| Variables | Frequency (%) |
|---|---|
| **Sex** | |
| Male | 368 (59.3) |
| Female | 253 (40.7) |
| **Age** | |
| >50 years | 79 (12.7) |
| >=50 years | 542 (87.3) |
| **Marital status** | |
| Single | 39 (6.3) |
| Married | 446 (71.8) |
| Widowed | 114 (18.4) |
| Divorced | 22 (3.5) |
| **Occupation** | |
| House wife | 158 (25.4) |
| Government employee | 163 (26.2) |
| Private employee | 120 (19.3) |
| **Business** | 180 (29) |
| Educational status | |
| No formal education | 122 (19.6) |
| Elementary school | 211 (34) |
| Secondary school | 119 (19.2) |
| Higher education | 169 (27.2) |
| **Monthly income** | |
| <500 Ethiopian Birr | 185 (29.8) |
| 501–2000 Ethiopian Birr | 79 (12.7) |
| >=2000 Ethiopian Birr | 357 (57.5) |

**Table 2. clinical characteristics and behavioral practice of hypertensive patients on follow up at selected public hospitals, Addis Ababa, Ethiopia (N = 632).**

| Variables | Frequency (%) |
| --- | --- |
| **Family history of hypertension** | |
| Yes | 142 (22.9) |
| No | 479 (77.1) |
| **Availability of sphygmomanometer at home** | |
| Yes | 238 (38.3) |
| No | 383 (61.7) |
| **Comorbidity** | |
| Yes | 275 (44.3) |
| No | 346 (55.7) |
| **Duration of disease** | |
| <5 years | 243 (39.1) |
| 5–10 years | 251 (40.4) |
| >10 years | 127 (20.5) |
| **Number of antihypertensive medication** | |
| 1 | 74 (11.9) |
| 2 | 362 (58.3) |
| >3 | 185 (29.8) |
| **Alcohol use** | |
| Yes | 97 (15.6) |
| No | 524 (84.4) |
| **Smoking** | |
| Yes | 53 (8.5) |
| No | 568 (91.5) |
| **Physical activity** | |
| Yes | 274 (44.1) |
| No | 347 (55.9) |
| **Salt intake** | |
| < 5 gram per day | 607 (97.7) |
| >= 5 gram per day | 14 (2.3) |
| **Medication adherence** | |
| Adhered | 433 (69.7) |
| Non-adhered | 188 (30.3) |
| **Physical activity adherence** | |
| Non -adhered | 539 (86.8) |
| Adhered | 82 (13.2) |
| **Uncontrolled Hypertension** | |
| No | 323 (52.0) |
| Yes | 298 (48.0) |

adherence, and physical activity. These variables were further evaluated in the multivariable model to account for potential confounding effects.

After adjustment for possible confounders on multivariable analysis, sex, educational status, occupational status, physical activity adherence, and drug adherence had significant associations with uncontrolled hypertension in multivariable analysis at 95% CI (p < 0.05).

Hypertensive patients with less than five years' duration of illness were 1.72 times more likely to have uncontrolled hypertension than those patients with greater than ten years (AOR = 1.72, 95% CI= (1.01–2.96)). Male hypertensive patients were 1.80 times more likely to have uncontrolled hypertension than female hypertensive patients (AOR = 1.80, 95% CI= (1.20–2.70)). Patients who did not adhere to physical activity were 2.48 times more likely to have uncontrolled hypertension than patients who were adhered to physical activity (AOR = 2.48, 95% CI= (1.39–4.41)). Housewife hypertensive patients were 62% less likely to have uncontrolled hypertension than business employee hypertensive patients (AOR = 0.38, 95% CI= (0.22–0.68)). Illiterate hypertensive patients were 2.45 times more likely to have uncontrolled hypertension than hypertensive patients with higher educational status (AOR = 2.45, 95% CI= (1.37–4.39)) (Table 3).

**Table 3. Multivariable and Bi-variable analysis for factors associated with uncontrolled hypertension among hypertensive patients on follow up at selected public hospitals, Addis Ababa, Ethiopia (N = 632).**

| Variables | Uncontrolled Hypertension Frequency (%) | | COR(95% CI) | AOR (95% CI |
|---|---|---|---|---|
| | Yes | No | | |
| **Sex** | | | | |
| Male | 193 (64.8) | 175 (54.2) | 1.55 (1.12–2.14) | 1.80 (1.20–2.70)** |
| Female | 105 (35.3) | 148 (45.8) | 1 | 1 |
| **Occupation** | | | | |
| House wife | 70 (23.5) | 88 (27.2) | 0.66 (0.43–1.02) | 0.38(0.22–0.68)*** |
| Government employee | 75 (25.2) | 88 (27.2) | 0.71 (0.46–1.09) | 0.65 (0.38–1.12) |
| Private employee | 55 (18.5) | 65 (20.1) | 0.70 (0.44–1.12) | 0.53(0.30–0.91)* |
| Business | 98 (32.9) | 82 (25.4 | 1 | 1 |
| **Educational status** | | | | |
| Illiterate | 62 (20.8) | 60 (18.6) | 1.69 (1.05–2.71) | 2.45 (1.37 - 4.39)** |
| Elementary school | 115 (38.6) | 96 (29.7) | 1.96(1.30–2.96) | 1.64 (0.95–2.83) |
| Secondary school | 57(19.1) | 62 (19.2) | 1.50 (0.93–2.42) | 1.48 (0.72–3.06) |
| Higher education | 64 (21.5) | 105 (32.5) | 1 | 1 |
| **Duration of hypertension** | | | | |
| <5 | 133 (44.6) | 110 (34.1) | 1.86 (1.20–2.88) | 1.70 (1.01–2.96)* |
| 5-10 | 115 (38.6) | 136 (42.1) | 1.30 (1.84–2.01) | 1.27 (0.75–2.15) |
| >10 | 50 (16.8) | 77(23.8) | 1 | 1 |
| **Number of anti-hypertensive medication** | | | | |
| 1 | 34 (11.4) | 48 (14.9) | 0.92(0.54–1.56) | 0.65(0.35–1.23) |
| 2 | 187 (62.8) | 175 (54.2) | 1.38(0.96–1.99) | 1.01(0.67–1.54) |
| >3 | 77 (25.8) | 100 (31) | 1 | 1 |
| **Medication adherence** | | | | |
| Non - adhered | 219 (73.5) | 214 (66.3) | 1.41 (1.00-1.99) | 1.36 (0.90 -2.06) |
| Adhered | 79 (26.5) | 109 (33.7) | 1 | 1 |
| **Physical activity** | | | | |
| Non – adhered | 275 (92.3) | 264 (81.7) | 2.67(1.60–4.45) | 2.48 (1.39-4.41)** |
| Adhered | 23 (7.7) | 59 (18.3) | 1 | 1 |

{P<0.05 =

*} {P<0.01 =

**} {P<=0.001 =

***}

## Discussion

In this study, the prevalence and risk factors for hypertension in patients were evaluated. This result demonstrated that 48% of hypertensive individuals had uncontrolled hypertension. This result was comparable with research done in Eastern Ethiopia 48% [20], Northern Ethiopia 48.6% [14], India 46.15% [21], and Saint Paul's Hospital 52.1% [15].

The finding of this study was lower than other studies done in Uganda 82.5% [18] and Bale zone public hospitals 56.7% [22]. However, the prevalence of this study was higher than the studies conducted in Egypt 33.2% [23] and Zewditu Memorial hospital 26.2% [24]. This inconsistency might be due to the proportion of non-adherence to their medication and exercise, socio demographic characteristics, study population, life style behavior and environmental factors.

In this study, uncontrolled hypertension was more common in male patients than in female patients. Women often adhere to most lifestyle modifications meant to manage hypertension more carefully than men, which might be one reasonable explanation [25]. Another reason might be that males forget they are using drugs because of the distraction provided by outside activities. Men tend to drink alcohol more often than women, which may also make it harder for them to follow treatment strategies [26].

Individuals with lower educational status showed a higher likelihood of uncontrolled hypertension than did individuals with higher educational status. A research done in Addis Ababa, Saint Paul's Hospital [15], and northern India [11] supports this conclusion. This might be explained by their lack of understanding about keeping their blood pressure at a reasonable level, their disregard for physical exercise, and their inability to perform frequent cheek raises. Another explanation might be non-adherence to medicine, which includes failing to take prescribed amounts, overdosing on prescriptions, and disregarding medical advice.

This study found that individuals who were not physically active were more likely to have uncontrolled hypertension than those who were. Studies from northern India [21], eastern Ethiopia [19], northern Ethiopia [14], and Ayder Comprehensive Hospital [27] are comparable to this one. One possible explanation is that exercise reduces blood pressure, enhances cardiac and kidney function, and avoids weight gain [28].

Another factor that has been substantially linked to uncontrolled hypertension is the length of the disease illness. Patients who have had their illness for less than five years are more likely to have uncontrolled hypertension than those who have had it for more than ten years. A research carried out at the Beadle General Hospital in southwest Ethiopia is comparable to this one [29]. This might be explained by the fact that when an illness progresses, aging also does, and comorbidity develops. Moreover, many chronic diseases impact endothelial cells, which disrupts blood vessel dilatation [30].

Peripheral vascular resistance and uncontrolled hypertension are indirectly caused by aging, which also promotes arterial stiffness and a loss of vasculature flexibility [14]. The other explanation might be the burden of taking too many pills as a result of comorbid conditions that make it difficult for patients to take their medications as prescribed. Additionally, this study demonstrated a substantial correlation between profession and uncontrolled hypertension, with business employers having lower rates of managed hypertension than housewives. This might be attributed to housewives spending the majority of their time taking care of their home, and finding time to take their medications as prescribed was difficult.

Due to its multi-center design, this research's results would be more broadly applicable and better reflect the study participants. Furthermore, the study used a sample size that is comparatively bigger, which is suitable for determining the prevalence and risk factors for uncontrolled hypertension. Although the research team has made every effort to obtain as trustworthy of data as possible, the results may not accurately reflect the truth due to the sensitive nature of some topics (such as alcohol use and cigarette smoking). While we standardized devices and measurement protocols, awareness of participation in the study may have influenced patients' BP readings (e.g., white-coat effect). To mitigate this, measurements were taken after rest, and results were withheld during data collection. Additionally, while mercury sphygmomanometers are gold-standard, periodic calibration and inter-operator variability could introduce minor measurement errors. Although our study provides valuable insights into hypertension management, we recognize that conducting the research in hospital settings rather than primary care facilities may introduce selection bias, as hospital patients often represent

more severe cases. We acknowledge this limitation while noting that hospital based recruitment was the most feasible approach within our current healthcare context in Addis Ababa.

## Conclusion

Uncontrolled hypertension was quite common among adult hypertensive individuals. Nearly half of the hypertensive individuals in this research had uncontrolled hypertension. Uncontrolled hypertension was substantially correlated with factors such as sex, employment, educational attainment, length of illness, adherence to physical exercise, and adherence to medication.

To address modifiable risk factors, healthcare providers should prioritize patient education on hypertension management, emphasizing medication adherence and regular physical activity. Additionally, targeted interventions for high-risk groups such as men, those with lower education, and those with shorter illness duration may improve outcomes.

Future research should explore causal relationships between these factors and uncontrolled hypertension through longitudinal or qualitative studies.

## Supporting information

**S1 Data. Minimal data set to support the finding of the study.**
(SAV)

## Acknowledgments

The authors acknowledge the data collectors, study participants, and the management bodies of the hospitals.

## Author contributions

**Conceptualization:** Abel Melese Teka, Tsega-Ab Abebaw Tekeba, Tadios Niguss Derese.

**Data curation:** Abel Melese Teka, Bemnet Yacobe Sayid, Tadios Niguss Derese.

**Formal analysis:** Abel Melese Teka, Dibora Yeshibelay Workineh, Tadios Niguss Derese.

**Investigation:** Abel Melese Teka, Dibora Yeshibelay Workineh, Eden Girma Beshah, Cheru Degf Tadesse, Tadios Niguss Derese.

**Methodology:** Abel Melese Teka, Tadios Niguss Derese.

**Project administration:** Tsega-Ab Abebaw Tekeba.

**Supervision:** Abel Melese Teka, Tsega-Ab Abebaw Tekeba, Bemnet Yacobe Sayid.

**Writing – original draft:** Abel Melese Teka, Abiye Assefa Berihun, Tadios Niguss Derese.

**Writing – review & editing:** Abel Melese Teka, Tsega-Ab Abebaw Tekeba, Dibora Yeshibelay Workineh, Abiye Assefa Berihun, Eden Girma Beshah, Cheru Degf Tadesse, Tadios Niguss Derese.

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
