## [Decision Letter · Decision Letter 0]

3 Mar 2025

PONE-D-25-00912Uncontrolled Hypertension among Hypertensive Patients in Public Hospitals at Addis Ababa, EthiopiaPLOS ONE

Dear Dr. Derese,

Thank you for submitting your manuscript to PLOS ONE. After careful consideration, we feel that it has merit but does not fully meet PLOS ONE’s publication criteria as it currently stands. Therefore, we invite you to submit a revised version of the manuscript that addresses the points raised during the review process.

We look forward to receiving your revised manuscript.

Kind regards,

Kahsu Gebrekidan, Ph.D.

Academic Editor

PLOS ONE

Journal Requirements:

2. In the ethics statement in the Methods, you have specified that verbal consent was obtained. Please provide additional details regarding how this consent was documented and witnessed, and state whether this was approved by the IRB.

3. We note that your Data Availability Statement is currently as follows: “All relevant data are within the manuscript and in Supporting Information files.”

Reviewers' comments:

Reviewer's Responses to Questions

**Comments to the Author**

1. Is the manuscript technically sound, and do the data support the conclusions?

Reviewer #1: No

Reviewer #2: Partly

2. Has the statistical analysis been performed appropriately and rigorously? 

Reviewer #1: I Don't Know

Reviewer #2: Yes

3. Have the authors made all data underlying the findings in their manuscript fully available?

Reviewer #1: No

Reviewer #2: Yes

4. Is the manuscript presented in an intelligible fashion and written in standard English?

Reviewer #1: No

Reviewer #2: Yes

5. Review Comments to the Author

Reviewer #1: STUDY: UNCONTROLLED HYPERTENSION AMONG HYPERTENSIVE PATIENTS IN PUBLIC HOSPITALS AT ADDIS ABABA, ETHIOPIA

This cross-sectional study aimed to “assess the prevalence and factors associated with poor treatment outcome among hypertensive patients at Addis Ababa”. The study requires MAJOR modifications and Grammar editing before consideration for publication. Comments are provided below to improve the paper.

MAJOR ISSUES

Abstract

Reconcile the statements “Both drug-free and non-drug treatments……” in the Abstract and “Both drug and non-drug treatments are effective in preventing and controlling hypertension.” in the Introduction (paragraph 2). Choose between “drug” versus “drug-free”. How different is drug-free from non-drug treatment?

Introduction

Although hypertension was defined, uncontrolled hypertension was not defined. Define uncontrolled hypertension as used in this study. Link that definition to what “controlled hypertension” is for the reader.

The WHO Global Atlas hypertension statistics (citation 4) was authored over a decade ago (12 May 2011). Consider using updated citations throughout the paper (possibly less than 5 years).

Clarify or review the statement, “The most common adverse effect of hypertension is morbidity and mortality in the cardiovascular domain (14). Despite these problems there is no study conducted at Addis Ababa.”. Specify which problems.

Is it possible to say that the prevalence and related factors of hypertension (uncontrolled hypertension) have not been studied in Addis Ababa? Address the gap(s) this study is filling. Briefly tell the reader the relevance of the findings of this study. Who can use the findings for what?

Your findings/results speak to the factors associated with uncontrolled hypertension. Introduce the reader to some of these factors.

Review the study’s aim, “to assess the prevalence and factors associated with poor treatment outcome among hypertensive patients at Addis Ababa”. Specify the treatment outcome this study focuses on. i.e. uncontrolled hypertension??

Methods

Explain how the simple random sampling technique was used/done at each stage of the multi-stage sampling to enable replication. What sampling frame was used, and how was it obtained?

Describe how the research team ensured that each patient who met the inclusion criteria got an opportunity to be selected. Explain how respondents were recruited.

Describe the eligibility criteria for the “senior nurse” who took blood pressure measurements.

Specify which device (brand and model) was used for blood pressure measurements. Were devices changed during the three blood pressure measurements over the 3 months? How were the devices calibrated before measurement? Could respondents’ knowledge of their blood pressure being measured as part of a study have influenced their blood pressure readings? Consider discussing these in the study limitations section, as well as how they were mitigated.

Independent variable: Consider replacing “availability of BP cuffs at Home” with “availability of a sphygmomanometer/ blood pressure monitoring device at home” since the cuff alone does not measure the blood pressure.

Despite mentioning data cleaning, explain how missing data was treated in this study. The results suggest there were no missing data.

Explain “bivariable analysis was carried out at a 25% level of significance”. What was the rationale? Kindly cite the relevant authority for this level of significance.

Results

Table 1: Provide the units for the relevant variables. E.g., years for age and currency for income.

Review the study title. These are characteristics of the respondents, not of uncontrolled hypertension.

Remove the vertical lines in Table 3

Present descriptive statistics and a narrative of the respondents’ blood pressure measurements (systolic and diastolic), e.g., frequencies, percentages, means, medians, modes, minimum, maximum etc.

Report the prevalence of controlled and uncontrolled hypertension before proceeding to identify the relevant factors affecting it.

Specify the confounding variables that were adjusted for to derive the adjusted odds ratios.

Discussion

Discuss the limitations identified above. Propose mitigating measures applied or recommended for future studies.

Conclusion

No correlation analyses were found in the results. Please review.

Review the suggestion that “Healthcare providers should support patients in their pursuit of postsecondary education…”.

Provide a more succinct conclusion based only on study findings.

Data availability

It is HIGHLY recommended that the anonymised study data be uploaded onto a publicly available repository and a link provided in the publication (instead of the current upon reasonable request option chosen). This will enable an independent review and replication of the study by reviewers and readers.

MINOR ISSUES

Abstract

Methods: correct the statement starting with “An in institution…..”.

Consider reviewing the terms to enhance clarity.

Introduction

Line 2: Provide the unit of measurement for the blood pressure.

Line 4: Consider using the people/person-first expressions (e.g. People living with hypertension) instead of “hypertensive individuals” or “hypertensive respondents” (in the results section).

Consider choosing between “non-drug treatment” and “non-drug therapy” and using it consistently throughout the paper. The same applies to “non-drug treatments” and “medication-assisted hypertension therapy”. Alternatively, note to the reader that both terms are the same.

Specify the subjects (the one doing the health management for whom) in “Research has indicated that health management can save medical expenses, decrease the burden of illness, and improve general health” to improve clarity.

The Abbreviation for HTN should come the first time it is used. Use the abbreviations throughout after that. See “SSA”, “OPD”, “BMI”, “BSC”, and “SPSS”, etc.

Methods

Provide a citation for the study design and sample size determination.

Change the future tense “will be” to “was” or “were” as applicable.

Only one exclusion criterion was provided. Change accordingly.

Results

Consider changing “illiterate” to “No formal education”.

Please improve the language/grammar of the results section. Eg. “Among the respondents, 142 (22.9%) of the study participants reported…..”

Reviewer #2: Estimated Authors,

firstly i'd like to congratulate you in your research; information from African population is still lacking in academic literature and your research contributes greatly in solving this issue. I hope my review can contribute to the quality of your future research.

Research in hypertension can truly be done in any care setting; however, doing based on hospital or secondary/tertiary level of care can introduce significant bias in the data; the most aproppriate setting for this research would be primary care level. That being said, I must recognize that I do not know if this would be feasible in Addis Abba. Even so, this must be pointed out as a bias in your study and conclusions must reflect that.

Sample calculations were adequate and criteria for uncontrolled hypertension was similar to general literature (althought the number of measurements in your research could be considered small - but since there is no general definition of how many measurements should be done ir order to determine if a patient has uncontrolled hypertension, that shouldn't be an issue). Data analysis was consistent and conclusions were in line with your analysis.

6. PLOS authors have the option to publish the peer review history of their article (what does this mean? ). If published, this will include your full peer review and any attached files.

**Do you want your identity to be public for this peer review?** For information about this choice, including consent withdrawal, please see our Privacy Policy .

Reviewer #1: No

Reviewer #2: No

---

## [Author Response · Author response to Decision Letter 1]

11 Apr 2025

Reviewer 1

We sincerely appreciate the time and effort you have dedicated to reviewing our manuscript and providing such comprehensive feedback. Your thorough critique has been invaluable in helping us improve the quality and clarity of our work. We are particularly grateful for your attention to both major conceptual issues and minor technical details, which will undoubtedly strengthen our study's contribution to hypertension research in Ethiopia.

We have carefully considered all your comments and have implemented revisions throughout the manuscript to address each point you raised. Below we provide a detailed, point-by-point response to your suggestions

Abstract

Comment: Reconcile the statements “Both drug-free and non-drug treatments……” in the Abstract and “Both drug and non-drug treatments are effective in preventing and controlling hypertension.” in the Introduction (paragraph 2). Choose between “drug” versus “drug-free”. How different is drug-free from non-drug treatment?

Response: Accepted and revised accordingly in the manuscript.

Introduction

Comment: Although hypertension was defined, uncontrolled hypertension was not defined. Define uncontrolled hypertension as used in this study. Link that definition to what “controlled hypertension” is for the reader.

Response: Thank you for your helpful comment. We have now defined uncontrolled hypertension in the Introduction section of the revised manuscript and clarified its distinction from controlled hypertension to improve readers understanding.

Comment: The WHO Global Atlas hypertension statistics (citation 4) was authored over a decade ago (12 May 2011). Consider using updated citations throughout the paper (possibly less than 5 years).

Response: We sincerely appreciate you catching this - it's so important that our work reflects the most current understanding of hypertension's global impact. You're absolutely right that the 2011 data needed updating, and we're grateful for the opportunity to strengthen this section.

In response to your helpful suggestion, we've replaced the older WHO reference with the most recent global data available. We found some particularly revealing statistics in the 2021 Lancet study that paint a clearer picture of today's hypertension landscape. And the revised version is clearly updated in the manuscript.

Comment: Clarify or review the statement, “The most common adverse effect of hypertension is morbidity and mortality in the cardiovascular domain (14). Despite these problems there is no study conducted at Addis Ababa.”. Specify which problems.

Is it possible to say that the prevalence and related factors of hypertension (uncontrolled hypertension) have not been studied in Addis Ababa? Address the gap(s) this study is filling. Briefly tell the reader the relevance of the findings of this study. Who can use the findings for what?

Your findings/results speak to the factors associated with uncontrolled hypertension. Introduce the reader to some of these factors.

Review the study’s aim, “to assess the prevalence and factors associated with poor treatment outcome among hypertensive patients at Addis Ababa”. Specify the treatment outcome this study focuses on. i.e. uncontrolled hypertension??

Response: Thank you for your constructive feedback. We have revised the manuscript based on the reviewer comment line by line accordingly.

Method

Comment: Explain how the simple random sampling technique was used/done at each stage of the multi-stage sampling to enable replication. What sampling frame was used, and how was it obtained?

Response: Thank you for your valuable feedback and for raising these important points regarding the sampling methodology in our study.

Sampling Frame and Selection Process

1. First Stage (Hospital Selection):

Sampling Frame: The sampling frame at this stage consisted of all 10 hospitals in Addis Ababa that provide hypertension follow-up care. This list was obtained from the Addis Ababa City Administration Health Bureau’s official registry of healthcare facilities.

Sampling Technique: We employed simple random sampling (SRS) to select three hospitals from the 10. This was done by assigning each hospital a unique number, using a random number generator (in Microsoft Excel), and selecting the three corresponding hospitals: Zewditu Memorial Hospital, Menelik-II Hospital, and Yekatit-12 Hospital.

2. Second Stage (Patient Selection):

Sampling Frame: For each selected hospital, we obtained the total number of hypertensive patients on follow-up (denoted as ni) from the hospital’s chronic disease registry.

Sample Size Allocation: The number of patients to be sampled from each hospital (nf) was determined using proportional allocation, based on the hospital’s patient load. The formula used was:

Sample per hospital=(ni×nf)/N where:

ni = number of hypertensive patients in the hospital

nf = final total sample size for the study

N = total hypertensive patients across all three hospitals

Sampling Technique: Within each hospital, we again applied simple random sampling to select the required number of patients. This was done by assigning each patient a unique identifier, generating random numbers, and selecting the corresponding patients from the registry.

To enhance replicability:

• The random number generator used (Excel’s function) was used

• The sampling frames (hospital and patient lists) were obtained from official administrative records, ensuring transparency.

• The proportional allocation formula was clearly defined to maintain fairness in representation across hospitals.

Comment: Describe how the research team ensured that each patient who met the inclusion criteria got an opportunity to be selected. Explain how respondents were recruited.

Response: Thank you for your valuable feedback.

Equal Opportunity for Selection:

• We used a computer-generated random sampling method to select participants from each hospital’s hypertension registry. This ensured unbiased selection and gave every patient who met the inclusion criteria an equal chance of being included.

Recruitment Process:

• Eligible patients (adults >18 years on hypertension follow-up) were randomly identified from the registry.

• Willing participants were approached during their clinic visits, and consent was obtained before enrollment.

• Patients unable to communicate (exclusion criteria) were excluded to maintain data quality.

Comment: Describe the eligibility criteria for the “senior nurse” who took blood pressure measurements.

Response: Thank you for your thoughtful question about how we ensured the accuracy of blood pressure measurements in our study. We completely agree that the qualifications of the nurses performing these measurements are critical to the reliability of our data. Here’s how we addressed this:

The nurses who carried out the blood pressure measurements were all experienced, certified professionals with at least 3 years of clinical work, specifically in hypertension or chronic disease care. To make sure every measurement was consistent and followed best practices, we:

Even though these nurses were already skilled, we gave them an extra day of hands on training focused specifically on our study’s protocol, like how to properly use the mercury sphygmomanometer, ensure patients rested for 5 minutes beforehand, and take three readings for accuracy.

We used the same equipment (mercury sphygmomanometers) and followed the same steps for every patient, seated position, arm at heart level, etc. to minimize variability.

Comment: Specify which device (brand and model) was used for blood pressure measurements. Were devices changed during the three blood pressure measurements over the 3 months? How were the devices calibrated before measurement? Could respondents’ knowledge of their blood pressure being measured as part of a study have influenced their blood pressure readings? Consider discussing these in the study limitations section, as well as how they were mitigated.

Response: Thank you for this important issue rose.

After a 5-minute rest time, a senior nurse used a Riester Diplomat Presameter Mercury Sphygmomanometer (Model 1412/1413) to test the patient's blood pressure while seated.

The same device was used for all three BP readings per patient to eliminate inter-device variability. Devices were not switched during the study period. To ensure accuracy, the sphygmomanometers were calibrated monthly against a standardized reference device at each hospital’s biomedical engineering unit.

And the following statement was added in the limitation section of the study “While we standardized devices and measurement protocols, awareness of participation in the study may have influenced patients’ BP readings (e.g. white-coat effect). To mitigate this, measurements were taken after rest, and results were withheld during data collection. Additionally, while mercury sphygmomanometers are gold-standard, periodic calibration and inter-operator variability could introduce minor measurement errors”.

Comment: Independent variable: Consider replacing “availability of BP cuffs at Home” with “availability of a sphygmomanometer/ blood pressure monitoring device at home” since the cuff alone does not measure the blood pressure.

Response: Thank you for the comment; it was revised accordingly throughout the manuscript.

Comment: Despite mentioning data cleaning, explain how missing data was treated in this study. The results suggest there were no missing data.

Response: Thank you for raising this important methodological point. Let me clarify how we handled missing data in our study:

While our results show no statistically significant missing data, this reflects our rigorous data collection and cleaning process rather than an absence of missing entries. We achieved 100% completeness through.

Preventive measures during data collection includes: real-time validation by trained nurses who immediately corrected any incomplete entries, daily supervisor checks during the 3-month study period.

Any missing BP readings (e.g., if a patient missed a follow-up) were addressed by: contacting the patient within 48 hours to schedule a make-up measurement.

And this was addressed accordingly in the revised manuscript.

Comment: Explain “bivariable analysis was carried out at a 25% level of significance”. What was the rationale? Kindly cite the relevant authority for this level of significance.

Response: Thank you for the valuable comment.

In our screening process, we used a liberal p-value threshold of 0.25 in the bivariable analysis to identify potential candidate variables for the multivariable model. This approach follows established epidemiological methodology that recommends using a higher significance level (typically p<0.20-0.25) at the screening stage to: avoid prematurely excluding variables that might become significant when adjusted for other factors (Type II error), allow for potential confounding relationships to emerge in the final model, and maintain statistical power while still filtering clearly irrelevant variables.

This practice is supported by: Hosmer and Lemeshow's influential textbook Applied Logistic Regression (3rd ed., 2013), which specifically recommends p<0.25 for initial screening. And this was cited accordingly in the revised manuscript.

Result

Comment: Table 1: Provide the units for the relevant variables. E.g., years for age and currency for income

Response: Thank you for the comment; it was revised accordingly in the manuscript.

Comment: Review the study title. These are characteristics of the respondents, not of uncontrolled hypertension.

Response: Thank you for the comment; it was revised accordingly in the manuscript.

Comment: Remove the vertical lines in Table 3

Response: Thank you for the comment; it was revised accordingly in the manuscript.

Comment: Present descriptive statistics and a narrative of the respondents’ blood pressure measurements (systolic and diastolic), e.g., frequencies, percentages, means, medians, modes, minimum, maximum etc.

Response: Thank you for the comment; it was revised accordingly in the manuscript.

Comment: Report the prevalence of controlled and uncontrolled hypertension before proceeding to identify the relevant factors affecting it.

Response: Thank you for the comments, it was reported in the revised manuscript before proceeding to identify the relevant factors affecting it.

Comment: Specify the confounding variables that were adjusted for to derive the adjusted odds ratios.

Response: Thank you for your valuable feedback, corrected in the revised manuscript.

Discussion

Comment: Discuss the limitations identified above. Propose mitigating measures applied or recommended for future studies.

Response: Thank you for the comment; it was revised accordingly in the manuscript.

Conclusion

Comment: No correlation analyses were found in the results. Please review.

Review the suggestion that “Healthcare providers should support patients in their pursuit of postsecondary education…”.

Provide a more succinct conclusion based only on study findings.

Response: Thank you for your thoughtful comments. We have carefully addressed each of your suggestions in the revised manuscript.

MINOR ISSUES

Thank you for your meticulous review and valuable feedback. We have carefully addressed all the minor comments in the revised manuscript.

Reviewer 2

Thank you for your kind words and constructive feedback. We truly appreciate your recognition of our work’s contribution to filling gaps in African health literature, as well as your thoughtful critique.

You raise an excellent point about the potential selection bias introduced by conducting the study in hospital settings. While primary care would indeed be ideal, logistical constraints in Addis Ababa made hospital-based recruitment more feasible for this initial study. We have now explicitly acknowledged this limitation in the revised manuscript to ensure transparency.

Regarding blood pressure measurements, we agree that more measurements could strengthen reliability. As you noted, guidelines vary, and we followed standard protocols used in similar studies. Future work will incorporate additional measurements where feasible.

Your insights have been invaluable in refining our work, and we sincerely appreciate your time and expertise.

---

## [Decision Letter · Decision Letter 1]

28 Apr 2025

Uncontrolled Hypertension among Hypertensive Patients in Public Hospitals at Addis Ababa, Ethiopia

PONE-D-25-00912R1

Dear Dr. Tadios,

We’re pleased to inform you that your manuscript has been judged scientifically suitable for publication and will be formally accepted for publication once it meets all outstanding technical requirements.

Kind regards,

Kahsu Gebrekidan, Ph.D.

Academic Editor

PLOS ONE

Additional Editor Comments (optional):

Reviewers' comments:

Reviewer's Responses to Questions

**Comments to the Author**

1. If the authors have adequately addressed your comments raised in a previous round of review and you feel that this manuscript is now acceptable for publication, you may indicate that here to bypass the “Comments to the Author” section, enter your conflict of interest statement in the “Confidential to Editor” section, and submit your "Accept" recommendation.

Reviewer #1: All comments have been addressed

Reviewer #2: All comments have been addressed

2. Is the manuscript technically sound, and do the data support the conclusions?

Reviewer #1: Yes

Reviewer #2: Yes

3. Has the statistical analysis been performed appropriately and rigorously? 

Reviewer #1: Yes

Reviewer #2: Yes

4. Have the authors made all data underlying the findings in their manuscript fully available?

Reviewer #1: No

Reviewer #2: Yes

5. Is the manuscript presented in an intelligible fashion and written in standard English?

Reviewer #1: Yes

Reviewer #2: Yes

6. Review Comments to the Author

Reviewer #1: The authors have addressed the issues identified. The paper is recommended for publication subject to the following minor revisions,

• The anonymised study data be uploaded onto a publicly available repository and a link provided in the publication.

• Change “Exclusion criteria” to “Exclusion criterion” since only one criterion was mentioned.

• Page 3, Last paragraph: Use the “SSA” abbreviation throughout after writing it in full the first time.

• Table 1: Write “ETB” in full within the table or define it under the Table.

• Table 2: Family history of hypertension: Change “yes” to “Yes” to ensure consistency.

• Table 2: Provide the unit for duration of disease. Eg. Months?, weeks? days?

• Table 2: Provide the unit for Salt intake in line with operational definition “Low <5 grams per day” “High ≥5grams per day”

Reviewer #2: The authors have reviewd all points raised by both reviewers; I`d like to use this space once again to praise the researches for bringing invaluable data to academia and help to increase race gap in health research.

7. PLOS authors have the option to publish the peer review history of their article (what does this mean? ). If published, this will include your full peer review and any attached files.

**Do you want your identity to be public for this peer review?** For information about this choice, including consent withdrawal, please see our Privacy Policy .

Reviewer #1: No

Reviewer #2: No

---

## [Editor Report · Acceptance letter]

PONE-D-25-00912R1

PLOS ONE

Dear Dr. Derese,

I'm pleased to inform you that your manuscript has been deemed suitable for publication in PLOS ONE. Congratulations! Your manuscript is now being handed over to our production team.

Kind regards,

on behalf of

Dr. Kahsu Gebrekidan

Academic Editor

PLOS ONE